# Gradient Sparsification for Communication-Efficient Distributed Optimization

**Jianqiao Wangni**
University of Pennsylvania
Tencent AI Lab
`wnjq@seas.upenn.edu`

**Jialei Wang**
Two Sigma Investments
`jialei.wang@twosigma.com`

**Ji Liu**
University of Rochester
Tencent AI Lab
`ji.liu.uwisc@gmail.com`

**Tong Zhang**
Tencent AI Lab
`tongzhang@tongzhang-ml.org`

## Abstract

Modern large-scale machine learning applications require stochastic optimization algorithms to be implemented on distributed computational architectures. A key bottleneck is the communication overhead for exchanging information such as stochastic gradients among different workers. In this paper, to reduce the communication cost, we propose a convex optimization formulation to minimize the coding length of stochastic gradients. The key idea is to randomly drop out coordinates of the stochastic gradient vectors and amplify the remaining coordinates appropriately to ensure the sparsified gradient to be unbiased. To solve the optimal sparsification efficiently, a simple and fast algorithm is proposed for an approximate solution, with a theoretical guarantee for sparseness. Experiments on $\ell_2$-regularized logistic regression, support vector machines and convolutional neural networks validate our sparsification approaches.

## 1 Introduction

Scaling stochastic optimization algorithms [26, 24, 14, 11] to distributed computational architectures [10, 17, 33] or multicore systems [23, 9, 19, 22] is a crucial problem for large-scale machine learning. In the synchronous stochastic gradient method, each worker processes a random minibatch of its training data, and then the local updates are synchronized by making an *All-Reduce* step, which aggregates stochastic gradients from all workers, and taking a *Broadcast* step that transmits the updated parameter vector back to all workers. The process is repeated until a certain convergence criterion is met. An important factor that may significantly slow down any optimization algorithm is the communication cost among workers. Even for the single machine multi-core setting, where the cores communicate with each other by reading and writing to a chunk of shared memory, conflicts of (memory access) resources may significantly degrade the efficiency. There are solutions to specific problems like mean estimation [29, 28], component analysis [20], clustering [6], sparse regression [16] and boosting [7]. Other existing works on distributed machine learning include two directions: 1) how to design communication efficient algorithms to reduce the round of communications among workers [37, 27, 12, 36], and 2) how to use large mini-batches without compromising the convergence speed [18, 31]. Several papers considered the problem of reducing the precision of gradient by using fewer bits to represent floating-point numbers [25, 2, 34, 8, 32] or only transmitting coordinates of large magnitudes[1, 21]. This problem has also drawn significant attention from theoretical perspectives about its communication complexity [30, 37, 3].

In this paper, we propose a novel approach to complement these methods above. Specifically, we sparsify stochastic gradients to reduce the communication cost, with minor increase in the number of iterations. The key idea behind our sparsification technique is to drop some coordinates of the stochastic gradient and appropriately amplify the remaining coordinates to ensure the unbiasedness of the sparsified stochastic gradient. The sparsification approach can significantly reduce the coding length of the stochastic gradient and only slightly increase the variance of the stochastic gradient. This paper proposes a convex formulation to achieve the optimal tradeoff of variance and sparsity: the optimal probabilities to sample coordinates can be obtained given any fixed variance budget. To solve this optimization within a linear time, several efficient algorithms are proposed to find approximately optimal solutions with sparsity guarantees. The proposed sparsification approach can be encapsulated seamlessly to many bench-mark stochastic optimization algorithms in machine learning, such as SGD [4], SVRG [14, 35], SAGA [11], and ADAM [15]. We conducted empirical studies to validate the proposed approach on $\ell_2$-regularized logistic regression, support vector machines, and convolutional neural networks on both synthetic and real-world data sets.

## 2    Algorithms

We consider the problem of sparsifying a stochastic gradient vector, and formulate it as a linear planning problem. Consider a training data set $\{x_n\}_{n=1}^N$ and $N$ loss functions $\{f_n\}_{n=1}^N$, each of which $f_n : \Omega \to \mathbb{R}$ depends on a training data point $x_n \in \Omega$. We use $w \in \mathbb{R}^d$ to denote the model parameter vector, and consider solving the following problem using stochastic optimization:

$$\min_w \quad f(w) := \frac{1}{N} \sum_{n=1}^N f_n(w), \quad w_{t+1} = w_t - \eta_t g_t(w_t), \tag{1}$$

where $t$ indicates the iterations and $\mathbb{E}\left[g_t(w)\right] = \nabla f(w)$ serves as an unbiased estimate for the true gradient $\nabla f(w_t)$. The following are two ways to choose $g_t$, like SGD [35, 4] and SVRG [14]

$$\text{SGD:} \quad g_t(w_t) = \nabla f_{n_t}(w_t), \qquad \text{SVRG:} \quad g_t(w_t) = \nabla f_{n_t}(w_t) - \nabla f_{n_t}(\widetilde{w}) + \nabla f(\widetilde{w}) \tag{2}$$

where $n_t$ is uniformly sampled from the data set and $\widetilde{w}$ is a reference point. The above algorithm implies that the convergence of SGD is significantly dominated by $\mathbb{E}\|g_t(w_t)\|^2$ or equivalently the variance of $g_t(w_t)$. It can be seen from the following simple derivation. Assume that the loss function $f(w)$ is $L$-smooth with respect to $w$, which means that for $\forall x, y \in \mathbb{R}^d, \|\nabla f(x) - \nabla f(y)\| \le L\|x-y\|$ (where $\|\cdot\|$ is the $\ell_2$-norm). Then the expected loss function is given by

$$\mathbb{E}\left[f(w_{t+1})\right] \le \mathbb{E}\left[f(w_t) + \nabla f(w_t)^\top (x_{t+1} - x_t) + \frac{L}{2}\|x_{t+1} - x_t\|^2\right] \tag{3}$$

$$=\mathbb{E}\left[f(w_t) - \eta_t \nabla f(w_t)^T g_t(w_t) + \frac{L}{2}\eta_t^2\|g_t(w_t)\|^2\right] = f(w_t) - \eta_t\|\nabla f(w_t)\|^2 + \frac{L}{2}\eta_t^2 \underbrace{\mathbb{E}\|g_t(w_t)\|^2}_{\text{variance}},$$

where the inequality is due to the Lipschitz property, and the second equality is due to the unbiased nature of the gradient $\mathbb{E}\left[g_t(w)\right] = \nabla f(w)$. So the magnitude of $\mathbb{E}(\|g_t(w_t)\|^2)$ or equivalently the variance of $g_t(w_t)$ will significantly affect the convergence efficiency.

Next we consider how to reduce the communication cost in distributed machine learning by using a sparsified gradient $g_t(w_t)$, denoted by $Q(g(w_t))$, such that $Q(g_t(w_t))$ is unbiased, and has a relatively small variance. In the following, to simplify notation, we denote the current stochastic gradient $g_t(w_t)$ by $g$ for short. Note that $g$ can be obtained either by SGD or SVRG. We also let $g_i$ be the $i$-th component of vector $g \in \mathbb{R}^d$: $g = [g_1, \ldots, g_d]$. We propose to randomly drop out the $i$-th coordinate by a probability of $1 - p_i$, which means that the coordinates remain non-zero with a probability of $p_i$ for each coordinate. Let $Z_i \in \{0, 1\}$ be a binary-valued random variable indicating whether the $i$-th coordinate is selected: $Z_i = 1$ with probability $p_i$ and $Z_i = 0$ with probability $1 - p_i$. Then, to make the resulting sparsified gradient vector $Q(g)$ unbiased, we amplify the non-zero coordinates, from $g_i$ to $g_i/p_i$. So the final sparsified vector is $Q(g)_i = Z_i(g_i/p_i)$. The whole protocol can be summarized as follows:

Gradients $g = [g_1, g_2, \cdots, g_d]$, Probabilities $p = [p_1, p_2, \cdots, p_d]$, Selectors $Z = [Z_1, Z_2, \cdots, Z_d]$,

$$\text{where } P(Z_i = 1) = p_i, \qquad \Longrightarrow \quad \text{Results } Q(g) = \left[Z_1\frac{g_1}{p_1}, Z_2\frac{g_2}{p_2}, \cdots, Z_d\frac{g_d}{p_d}\right] \tag{4}$$

We note that if $g$ is an unbiased estimate of the gradient, then $Q(g)$ is also an unbiased estimate of the gradient since $\mathbb{E}\left[Q(g)_i\right] = p_i \times \frac{g_i}{p_i} + (1 - p_i) \times 0 = g_i$.

In distributed machine learning, each worker calculates gradient $g$ and transmits it to the master node or the parameter server for an update. We use an index $m$ to indicate a node, and assume there are total $M$ nodes. The gradient sparsification method can be used with a synchronous distributed stochastic optimization algorithm in Algorithm 1. Asynchronous algorithms can also be used with our technique in a similar fashion.

---

**Algorithm 1** A synchronous distributed optimization algorithm

1: Initialize the clock $t = 0$ and initialize the weight $w_0$.
2: **repeat**
3:     Each worker $m$ calculates local gradient $g^m(w_t)$ and the probability vector $p^m$.
4:     Sparsify the gradients $Q(g^m(w_t))$ and take an *All-Reduce* step $v_t = \frac{1}{M}\sum_{m=1}^{M} Q(g^m(w_t))$.
5:     Broadcast the average gradient $v_t$ and take a descent step $w_{t+1} = w_t - \eta_t v_t$ on all workers.
6: **until** convergence or the number of iteration reaches the maximum setting.

---

Our method could be combined with other methods which are orthogonal to us, like only transmitting large coordinates and accumulating the gradient residual which might be transmitted in the next step [1, 21]. Advanced quantization and coding strategy from [2] can be used for transmitting valid coordinates of our method. In addition, this method concords with [29] for the mean estimation problem on distributed data, with a statistical guarantee under skewness.

## 2.1 Mathematical formulation

Although the gradient sparsification technique can reduce communication cost, it increases the variance of the gradient vector, which might slow down the convergence rate. In the following section, we will investigate how to find the optimal tradeoff between sparsity and variance for the sparsification technique. In particular, we consider how to find out the optimal sparsification strategy, given a budget of maximal variance. First, note that the variance of $Q(g)$ can be bounded by

$$\mathbb{E}\sum_{i=1}^{d}[Q(g)_i^2] = \sum_{i=1}^{d}\left[\frac{g_i^2}{p_i^2} \times p_i + 0 \times (1 - p_i)\right] = \sum_{i=1}^{d}\frac{g_i^2}{p_i}. \tag{5}$$

In addition, the expected sparsity of $Q(g_i)$ is given by $\mathbb{E}\left[\|Q(g)\|_0\right] = \sum_{i=1}^{d} p_i$. In this paper, we try to balance these two factors (sparsity and variance) by formulating it as a linear planning problem as follows:

$$\min_{p}\quad \sum_{i=1}^{d} p_i \quad \text{s.t.} \quad \sum_{i=1}^{d}\frac{g_i^2}{p_i} \leq (1 + \epsilon)\sum_{i=1}^{d} g_i^2, \tag{6}$$

where $0 < p_i \leq 1, \forall i \in [d]$, and $\epsilon$ is a factor that controls the variance increase of the stochastic gradient $g$. This leads to an optimal strategy for sparsification given an upper bound on the variance. The following proposition provides a closed-form solution for problem (6).

**Proposition 1.** *The solution to the optimal sparsification problem* (6) *is a probability vector $p$ such that $p_i = \min(\lambda|g_i|, 1), \forall i \in [d]$, where $\lambda > 0$ is a constant only depending on $g$ and $\epsilon$.*

*Proof.* By introducing Lagrange multipliers $\lambda$ and $\mu_i$, we know that the solution of (6) is given by the solution of the following objective:

$$\min_{p}\max_{\lambda}\max_{\mu} L(p_i, \lambda, \mu_i) = \sum_{i=1}^{d} p_i + \lambda^2\left(\sum_{i=1}^{d}\frac{g_i^2}{p_i} - (1 + \epsilon)\sum_{i=1}^{d} g_i^2\right) + \sum_{i=1}^{d}\mu_i(p_i - 1). \tag{7}$$

Consider the KKT conditions of the above formulation, by stationarity with respect to $p_i$ we have:

$$1 - \lambda^2\frac{g_i^2}{p_i^2} + \mu_i = 0, \quad \forall i \in [d]. \tag{8}$$

Note that we have to permit $p_i = 0$ for KKT condition to apply. Combined with the complementary slackness condition that guarantees $\mu_i(p_i - 1) = 0, \forall i \in [d]$, we know that $p_i = 1$ for $\mu_i \neq 0$, and $p_i = \lambda|g_i|$ for $\mu_i = 0$. This tells us that for several coordinates the probability of keeping the value is 1 (when $\mu_i \neq 0$), and for other coordinates the probability of keeping the value is proportional to the magnitude of the gradient $g_i$. Also, by simple reasoning we know that if $|g_i| \geq |g_j|$ then $|p_i| \geq |p_j|$ (otherwise we simply switch $p_i$ and $p_j$ and get a sparser result). Therefore there is a dominating set of coordinates $S$ with $p_j = 1, \forall j \in S$, and it must be the set of $|g_j|$ with the largest absolute magnitudes. Suppose this set has a size of $|S| = k$ ($0 \leq k \leq d$) and denote by $g_{(1)}, g_{(2)}, ..., g_{(d)}$ the elements of $g$ ordered by their magnitudes (for the largest to the smallest), we have $p_i = 1$ for $i \leq k$, and $p_i = \lambda|g_i|$ for $i > k$. $\qquad\square$

## 2.2 Sparsification algorithms

In this section, we propose two algorithms for efficiently calculating the optimal probability vector $p$ in Proposition 1. Since $\lambda > 0$, by the complementary slackness condition, we have

$$\sum_{i=1}^{d} \frac{g_i^2}{p_i} - (1+\epsilon)\sum_{i=1}^{d} g_i^2 = \sum_{i=1}^{k} g_{(i)}^2 + \sum_{i=k+1}^{d} \frac{|g_{(i)}|}{\lambda} - (1+\epsilon)\sum_{i=1}^{d} g_i^2 = 0. \qquad (9)$$

This further implies

$$\lambda = (\epsilon \sum_{i=1}^{d} g_i^2 + \sum_{i=k+1}^{d} g_{(i)}^2)^{-1} (\sum_{i=k+1}^{d} |g_{(i)}|), \qquad (10)$$

then we used the constraint $\lambda|g_{(k+1)}| \leq 1$ and get

$$|g_{(k+1)}| \left( \sum_{i=k+1}^{d} |g_{(i)}| \right) \leq \epsilon \sum_{i=1}^{d} g_i^2 + \sum_{i=k+1}^{d} g_{(i)}^2. \qquad (11)$$

It follows that we should find the smallest $k$ which satisfies the above inequality. Based on the above reasoning, we get the following closed-form solution for $p_i$ in Algorithm 2.

---
**Algorithm 2** Closed-form solution
---
1: Find the smallest $k$ such that the second inequality of (10) is true, and let $S_k$ be the set of coordinates with top $k$ largest magnitude of $|g_i|$.
2: Set the probability vector $p$ by

$$p_i = \begin{cases} 1, & \text{if } i \in S_k \\ (\epsilon\sum_{j=1}^{d} g_{(j)}^2 + \sum_{j=k+1}^{d} g_{(j)}^2)^{-1}|g_i| \left(\sum_{j=k+1}^{d} |g_{(j)}|\right), & \text{if } i \notin S_k. \end{cases}$$

---

In practice, Algorithm 2 requires partial sorting of the gradient magnitude values to find $S_k$, which could be computationally expensive. Therefore we developed a greedy algorithm for approximately solving the problem. We pre-define a sparsity parameter $\kappa \in (0,1)$, which implies that we aim to find $p_i$ that satisfies $\sum_i p_i/d \approx \kappa$. Loosely speaking, we want to initially set $\widetilde{p}_i = \kappa d|g_i|/\sum_i |g_i|$, which sums to $\sum_i \widetilde{p}_i = \kappa d$, meeting our requirement on $\kappa$. However, by the truncation operation $p_i = \min(\widetilde{p}_i, 1)$, the expected nonzero density will be less than $\kappa$. Now, we can use an iterative procedure that in the next iteration, we fix the set of $\{p_i : p_i = 1\}$ and scale the remaining values, as summarized in Algorithm 3. This algorithm is much easier to implement, and computationally more efficient on parallel computing architecture. Since the operations mainly consist of accumulations, multiplications and minimizations, they can be easily accelerated on graphic processing units (GPU) or other hardware supporting *single instruction multiple data (SIMD)*.

## 2.3 Coding strategy

Once we have computed a sparsified gradient vector $Q(g)$, we need to pack the resulting vector into a message for transmission. Here we apply a hybrid strategy for encoding $Q(g)$. Suppose that

**Algorithm 3** Greedy algorithm

1: **Input** $g \in \mathbb{R}^d$, $\kappa \in (0,1)$. Initialize $p^0 \in \mathbb{R}^d$, $j = 0$. Set $p_i^0 = \min(\kappa d |g_i| / \sum_i |g_i|, 1)$ for all $i$.
2: **repeat**
3:     Identify an active set $\mathcal{I} = \{1 \leq i \leq D | p_i^j \neq 1\}$ and compute $c = (\kappa d - d + |\mathcal{I}|) / \sum_{i \in \mathcal{I}} p_i^j$.
4:     Recalibrate the values by $p_i^{j+1} = \min(c p_i^j, 1)$. $j = j + 1$.
5: **until** If $c \leq 1$ or $j$ reaches the maximum iterations. Return $p = p^j$.

computers represent a floating-point scalar using $b$ bits, with negligible loss in precision. We use two vectors $Q_A(g)$ and $Q_B(g)$ for representing non-zero coordinates, one for coordinates $i \in S_k$, and the other for coordinates $i \notin S_k$. The vector $Q_A(g)$ represents $\{g_i : i \in S_k\}$, where each item of $Q_A(g)$ needs $\log d$ bits to represent the coordinates and $b$ bits for the value $g_i / p_i$. The vector $Q_B(g)$ represents $\{g_i : i \notin S_k\}$, since in this case, we have $p_i = \lambda |g_i|$, we have for all $i \notin S_k$ the quantized value $Q(g_i) = g_i / p_i = \text{sign}(g_i) / \lambda$. Therefore to represent $Q_B(g)$, we only need one floating-point scalar $1/\lambda$, plus the non-zero coordinates $i$ and its sign $\text{sign}(g_i)$. Here we give an example about the format,

$$Q(g): \quad \left[ \frac{g_1}{p_1}, 0, 0, \frac{g_4}{p_4}, \frac{g_5}{p_5}, \frac{g_6}{p_6}, \cdots, 0 \right], \quad \text{where } g_1, g_5 \in S_k, g_4 < 0, g_6 > 0,$$

$$Q_A(g): \quad \left[ 1, \frac{g_1}{p_1}, 5, \frac{g_5}{p_5} \cdots, 0 \right], \qquad Q_B(g): \quad [4, -1/\lambda, 6, 1/\lambda, \cdots]. \tag{12}$$

where $i = 1, 5 \in S_k, i = 4, 6 \notin S_k, g_4 < 0, g_6 > 0$. Moreover, we can also represent the indices of $A$ and vector $Q_B(g)$ using a dense vector of $\widetilde{q} \in \{0, \pm 1, 2\}^d$, where each component $\widetilde{q}_i$ is defined as $Q(g_i) = \lambda Q(g_i)$ when $i \notin S_k$ and $\widetilde{q}_i = 2$ if $i \in S_k$. Using the standard entropy coding, we know that $\widetilde{q}$ requires at most $\sum_{\ell=-1}^{2} d_\ell \log_2(d/d_\ell) \leq 2d$ bits to represent.

## 3   Theoretical guarantees on sparsity

In this section we analyze the expected sparsity of $Q(g)$, which equals to $\sum_{i=1}^{d} p_i$. In particular we show when the distribution of gradient magnitude values is highly skewed, there is a significant gain in applying the proposed sparsification strategy. First, we define the following notion of approximate sparsity on the magnitude at each coordinate of $g$:

**Definition 2.** *A vector $g \in \mathbb{R}^d$ is $(\rho, s)$-approximately sparse if there exists a subset $S \subset [d]$ such that $|S| = s$ and $\|g_{S^c}\|_1 \leq \rho \|g_S\|_1$, where $S^c$ is the complement of $S$.*

The notion of $(\rho, s)$-approximately sparsity is inspired by the restricted eigenvalue condition used in high-dimensional statistics [5]. $(\rho, s)$-approximately sparsity measures how well the signal of a vector is concentrated on a small subset of the coordinates of size $s$. As we will see later, the quantity $(1 + \rho)s$ plays an important role in establishing the expected sparsity bound. Note that we can always take $s = d$ and $\rho = 0$ so that $(\rho, s)$ satisfies the above definition with $(1 + \rho)s \leq d$. If the distribution of magnitude values in $g$ is highly skewed, we would expect the existence of $(\rho, s)$ such that $(1 + \rho)s \ll d$. For example when $g$ is exactly $s$-sparse, we can choose $\rho = 0$ and the quantity $(1 + \rho)s$ reduces to $s$ which can be significantly smaller than $d$.

**Lemma 3.** *If the gradient $g \in \mathbb{R}^d$ of the loss function is $(\rho, s)$-approximately sparse as in Definition 2. Then we can find a sparsification $Q(g)$ with $\epsilon = \rho$ in (6) (that is, the variance of $Q(g)$ is increased by a factor of no more than $1 + \rho$), and the expected sparsity of $Q(g)$ can be upper bounded by $\mathbb{E}[\|Q(g)\|_0] \leq (1 + \rho)s$.*

*Proof.* Based on Definition 2, we can choose $\epsilon = \rho$ and $S_k = S$ that satisfies (10), thus

$$\mathbb{E}[\|Q(g)\|_0] = \sum_{i=1}^{d} p_i = \sum_{i \in S_k} p_i + \sum_{i \notin S_k} p_i = s + \sum_{i \notin S_k} \frac{|g_i|(\sum_{j=k+1}^{d} |g_{(j)}|)}{\epsilon \sum_{j=1}^{k} g_{(j)}^2 + (1 + \epsilon) \sum_{j=k+1}^{d} g_{(j)}^2}$$

$$= s + \frac{\|g_{S_k^c}\|_1^2}{\rho \|g_{S_k}\|_2^2 + (1 + \rho) \|g_{S_k^c}\|_2^2} \leq s + \frac{\rho^2 s \|g_{S_k}\|_2^2}{\rho \|g_{S_k}\|_2^2 + (1 + \rho) \|g_{S_k^c}\|_2^2} \leq (1 + \rho)s, \tag{13}$$

which completes the proof. □

**Remark 1.** *Lemma 3 indicates that the variance after sparsification only increases by a factor of $(1+\rho)$, while in expectation we only need to communicate a $(1+\rho)s$-sparse vector after sparsification. In order to achieve the same optimization accuracy, we may need to increase the number of iterations by a factor of up to $(1 + \rho)$, and the overall number of floating-point numbers communicated is reduced by a factor of up to $(1 + \rho)^2 s/d$.*

Above lemma shows the number of floating-point numbers needed to communicate per iteration is reduced by the proposed sparsification strategy. As shown in Section 2.3, we only need to use one floating-point number to encode the gradient values in $S_k^c$, so there is a further reduction in communication when considering the total number of bits transmitted, this is characterized by the Theorem below. The details of proof are put in a full version (`https://arxiv.org/abs/1710.09854`) of this paper.

**Theorem 4.** *If the gradient $g \in \mathbb{R}^d$ of the loss function is $(\rho, s)$-approximately sparse as in Definition 2, and a floating-point number costs $b$ bits, then the coding length of $Q(g)$ in Lemma 3 can be bounded by $s(b + \log_2 d) + \min(\rho s \log_2 d, d) + b$.*

**Remark 2.** *The coding length of the original gradient vector $g$ is $db$, by considering the slightly increased number of iterations to reach the same optimization accuracy, the total communication cost is reduced by a factor of at least $(1 + \rho)((s + 1)b + \log_2 d)/db$.*

## 4 Experiments

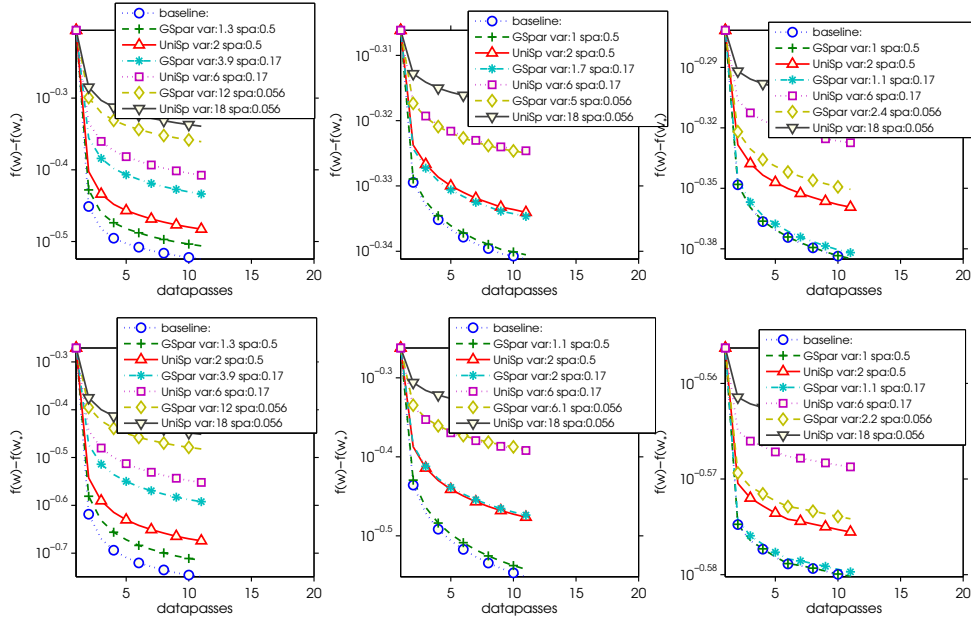

Figure 1: SGD type comparison between gradient sparsification (GSpar) with random sparsification with uniform sampling (UniSp).

In this section we conduct experiments to validate the effectiveness and efficiency of the proposed sparsification technique. We use $\ell_2$-regularized logistic regression as an example for convex problems, and take convolutional neural networks as an example for non-convex problems. The sparsification technique shows strong improvement over the uniform sampling approach as a baseline, the iteration complexity is only slightly increased as we strongly reduce the communication costs. Moreover, we also conduct asynchronous parallel experiments on the shared memory architecture. In particular, our experiments show that the proposed sparsification technique significantly reduces the conflicts among multiple threads and dramatically improves the performance. In all experiments, the probability vector $p$ is calculated by Algorithm 3 and set the maximum iterations to be 2, which generates good enough high-quality approximation of the optimal $p$ vector.

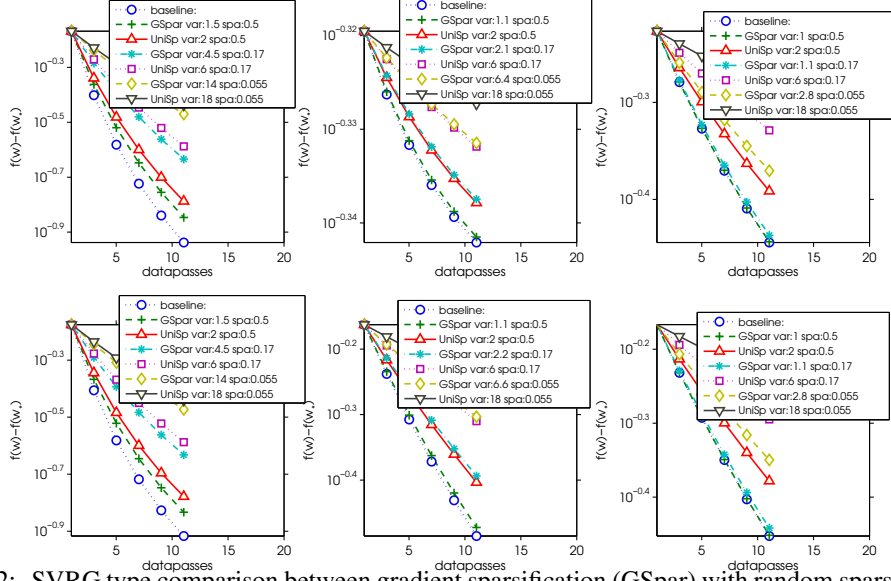

Figure 2: SVRG type comparison between gradient sparsification (GSpar) with random sparsification with uniform sampling (UniSp)

We first validate the sparsification technique on the $\ell_2$-regularized logistic regression problem using SGD and SVRG respectively: $f(w) = \frac{1}{N} \sum_n \log_2 \left(1 + \exp(-a_n^\top w b_n)\right) + \lambda_2 \|w\|_2^2$, where $a_n \in \mathbb{R}^d$, $b_n \in \{-1, 1\}$. The experiments are conducted on synthetic data for the convenience to control the data sparsity. The mini-batch size is set to be 8 by default unless otherwise specified. We simulated with $M = 4$ machines, where one machine is both a worker and the master that aggregates stochastic gradients received from other workers. We compare our algorithm with a uniform sampling method as baseline, where each element of the probability vector is set to be $p_i = \kappa$, and a similar sparsification follows to apply. In this method, the sparsified vector has a nonzero density of $\kappa$ in expectation. The data set $\{a_n\}_{n=1}^N$ is generated as follows

dense data: $\quad \bar{a}_{ni} \sim \mathcal{N}(0, 1), \quad \forall i \in [d], n \in [N],$ sparsify: $\quad \bar{B} \sim \text{Uniform}[0, 1]^d, \quad \bar{B}_i \leftarrow C_1 \bar{B}_i,$

if: $\quad \bar{B}_i \leq C_2, \quad \forall i \in [d], \quad a_n \leftarrow \bar{a}_n \odot \bar{B},$ label: $\quad \bar{w} \sim \mathcal{N}(0, I), \quad b_n \leftarrow \text{sign}(\bar{a}_n^\top \bar{w})$

where $\odot$ is the element-wise multiplication. In the equations above, the first step is a standard data sampling procedure from a multivariate Gaussian distribution; the second step generates a magnitude vector $\bar{B}$, which is later sparsified by decreasing elements that are smaller than a threshold $C_2$ by a factor of $C_1$; the third line describes the application of magnitude vectors on the dataset; and the fourth line generates a weight vector $\bar{w}$, and labels $y_n$, based on the signs of multiplications of data and the weights. We should note that the parameters $C_1$ and $C_2$ give us a easier way to control the sparsity of data points and the gradients: the smaller these two constants are, the sparser the gradients are. The gradient of linear models on the dataset should be expected to be $\left((1 - C_2)d, C_2 \frac{C_1}{C_1+2}\right)$-approximately sparse, and the gradient of regularization needs not to be communicated. We set the dataset of size $N = 1024$, dimension $d = 2048$. The step sizes are fine-tuned on each case, and in our findings, the empirically optimal step size is inversely related to the gradient variance as the theoretical analysis.

In Figures 1 and 2, from the top row to the bottom row, the $\ell_2$-regularization parameter $\lambda$ is set to $1/(10N)$, $1/N$. And in each row, from the first column to the last column, $C_2$ is set to $4^{-1}, 4^{-2}, 4^{-3}$. In these figures, our algorithm is denoted by '*GSpar*', and the uniform sampling method is denoted by '*UniSp*', and the SGD/SVRG algorithm with non-sparsified communication is denoted by '*baseline*', indicating the original distributed optimization algorithm. The $x$-axis shows the number of data passes, and the $y$-axis draws the suboptimality of the objective function $(f(w_t) - \min_w f(w))$. For the experiments, we report the sparsified-gradient SGD variance as the notation '*var*' in Figure 1. And '*spa*' in all figures represents the nonzero density $\kappa$ in Algorithm 3. We observe that the theoretical complexity reduction against the baseline in terms of the communication rounds, which can be inferred by *var* $\times$ *spa*, from the labels in Figures 1 to 2, where $C_1 = 0.9$, and the rest of the figures are put in the full version due to the limited space.

From Figure 1, we observe that results on sparser data yield smaller gradient variance than results on denser data. Compared to uniform sampling, our algorithm generates gradients with less variance, and converges much faster. This observation is consistent with the objective of our algorithm, which is to minimize gradient variance given a certain sparsity. The convergence slowed down linearly w.r.t. the increase of variance. The results on SVRG show better speed up — although our algorithm increases the variance of gradients, the convergence rate degrades only slightly.

We compared the gradient sparsification method with the quantized stochastic gradient descent (QSGD) algorithm in [2]. The results are shown in Figures 4. The data are generated as previous, with both strong and weak sparsity settings. From the top row to the bottom row, the $\ell_2$-regularization parameter $\lambda$ is set to $1/(10N)$, $1/N$. And in each row, from the first column to the last column, $C_2$ is set to $4^{-1}$, $4^{-2}$. The step sizes are set to be the same for both methods for a fair comparison after fine-tuning. In this comparison, we use the overall communication coding length of each algorithm, and note the length in $x$-axis. For QSGD, the communication cost per element is linearly related to $b$, which refers to the bits of floating-point num-

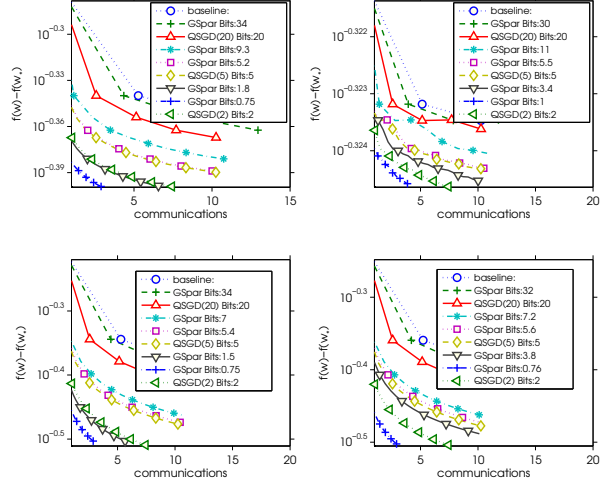

Figure 3: Comparison of the sparisified-SGD with QSGD.

ber. QSGD($b$) denotes QSGD algorithm with bit number $b$ in these figures, and the average bits required to represent per element is on the labels. We also tried to compare with the gradient residual accumulation approaches [1], which unfortunately failed on our experiments, since the gradient is relatively sparse so that lots of small coordinates could be delayed infinitely, resulting in a large gradient bias to cause the divergence on convex problems. From Figures 4, we observe that the proposed sparsification approach is at least comparable to QSGD, and significantly outperforms QSGD when the gradient sparsity is stronger; and this concords with our analysis on the gradient approximate sparsity encouraging faster speed up.

## 4.1 Experiments on deep learning

This section conducts experiments on non-convex problems. We consider the convolutional neural networks (CNN) on the CIFAR-10 dataset with different settings. Generally, the networks consist of three convolutional layers ($3 \times 3$), two pooling layers ($2 \times 2$), and one $256$ dimensional fully connected layer. Each convolution layer is followed by a batch-normalization layer. The channels of each convolutional layer is set to $\{24, 32, 48, 64\}$. We use the ADAM optimization algorithm [15], and the initial step size is set to $0.02$.

In Figure 4.1, we plot the objective function against the computational complexity measured by the number

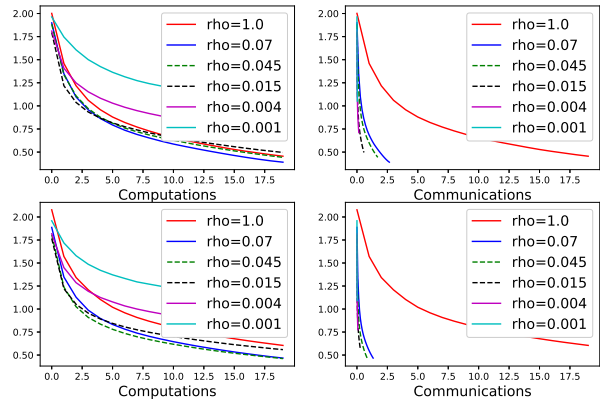

Figure 4: Comparison of 3-layer CNN of channels of 64 (top) and 48 (bottom) on CIFAR-10. (Y-axis: $f(w_t)$.)

of epochs (1 epoch is equal to 1 pass of all training samples). We also plot the convergence with respect to the communication cost, which is the product of computations and the sparsification pa-

rameter $\kappa$. The experiments on each setting are repeated 4 times and we report the average objective function values. The results show that for this non-convex problem, the gradient sparsification slows down the training efficiency only slightly. In particular, the optimization algorithm converges even when the sparsity ratio is about $\kappa = 0.004$, and the communication cost is significantly reduced in this setting. This experiments also show that the optimization of neural networks is less sensitive to gradient noises, and the noises within a certain range may even help the algorithm to avoid bad local minimums [13].

## 4.2 Experiments on asynchronous parallel SGD

In this section, we study parallel implementations of SGD on the single-machine multi-core architecture. We employ the support vector machine for binary classification, where the loss function is $f(w) = \frac{1}{N}\sum_n \max(1 - a_n^\top w b_n, 0) + \lambda_2 \|w\|_2^2$, $a_n \in \mathbb{R}^d$, $b_n \in \{-1, 1\}$. We implemented shared memory multi-thread SGD, where each thread employs a locked read, which may block other threads' writing to the same coordinate. We implement a multi-thread algorithm with locks which are implemented using *compare-and-swap* operations. To improve the speed of the algorithm, we also employ several engineering tricks. First, we observe that $\forall p_i < 1, \quad g_i/p_i = \text{sign}(g_i)/\lambda$ from Proposition 1, therefore we only need to assign constant values to these variables, without applying floating-point division operations. Another costly operation is the pseudo-random number generation in the sampling procedure; therefore we generate a large array of pseudo-random numbers in $[0, 1]$, and iteratively read the numbers during training without calling a random number generating function. The data are generated by first generating dense data, sparsifying them and generating the corresponding labels:

$$\bar{a}_{ni} \sim \mathcal{N}(0, 1), \forall i \in [d], n \in [N], \bar{w} \sim \text{Uniform}[-0.5, 0.5]^d, \bar{B} \sim \text{Uniform}[0, 1]^d,$$

$$\bar{B}_i \leftarrow C_1 \bar{B}_i, \text{if:} \bar{B}_i \leq C_2, \forall i \in [d], a_n \leftarrow \bar{a}_n \odot \bar{B}, \quad b_n \leftarrow \text{sign}(x_n^\top \bar{w} + \sigma), \text{where } \sigma \sim \mathcal{N}(0, 1).$$

We set the dataset of size $N = 51200$, dimension $d = 256$, also set $C_1 = 0.01$ and $C_2 = 0.9$.

The regularization parameter $\lambda_2$ is denoted by $reg$, the number of threads is denoted by $W$(workers), and the learning rate is denoted by $lrt$. The number of workers is set to 16 or 32, the regularization parameter is set to $\{0.5, 0.1, 0.05\}$, and the learning rate is chosen from $\{0.5, 0.25, 0.05, 0.25\}$. The convergence of objective value against running time (milliseconds) is plotted in Figure 4.2, and the rest of figures are put in the full version.

From Figure 4.2, we can observe that using gradient sparsification, the conflicts among multiple threads for reading and writing the same coordinate are significantly reduced. Therefore the training speed is significantly faster. By comparing with other set-

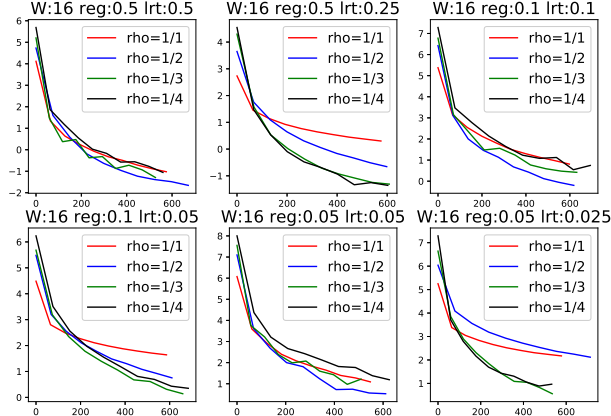

Figure 5: Loss functions by a multi-thread SVM. X-axis: time in milliseconds, Y-axis: $\log_2(f(w_t))$.

tings, we also observe that the sparsification technique works better at the case when more threads are available, since the more threads, the more frequently the lock conflicts occur.

## 5 Conclusions

In this paper, we propose a gradient sparsification technique to reduce the communication cost for large-scale distributed machine learning. We propose a convex optimization formulation to minimize the coding length of stochastic gradients given the variance budget that monotonically depends on the computational complexity, with efficient algorithms and a theoretical guarantee. Comprehensive experiments on distributed and parallel optimization of multiple models proved our algorithm can effectively reduce the communication cost during training or reduce conflicts among multiple threads.

## Acknowledgments

Ji Liu is in part supported by NSF CCF1718513, IBM faculty award, and NEC fellowship.

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
