[Reviews · NeurIPS 2018]

Reviewer 1



This paper focuses on large-scale machine learning tasks in a distributed computing setup with the objective of making the underlying optimization procedures communication efficient. In particular, the paper considers the problem of reducing the overall communication during the collection of the stochastic gradient of the objective function at hand. The authors propose a sparsification approach where, during an iteration of the optimization procedure, each coordinate of the stochastic gradient is independently replaced with zero value with a certain probability. The non-zero coordinates are then properly scaled to ensure that the sparsified gradient remains an unbiased estimate of the true gradient. Now the (scaled) sparse gradient vector can be encoded up to a desired level of precision. The randomized sparsification of the (original) stochastic gradient generates another stochastic gradient with higher variance, which may lead to poor performance in term of the convergence rate of the optimization procedure. Keeping this in mind, the authors tune their sparsification procedure to ensure the variance of the resulting stochastic gradient is not too high. This allows them to simultaneously ensure both good convergence rate and communication-efficiency. In particular, the authors provably show that their approach has very good performance when the original stochastic gradient is sparse or approximately sparse. The proposed scheme is practical and can be combined with various other existing approaches to reduce the communication overhead in a distributed computing setting. The authors conduct extensive experiments involving logistic regression and neural network training to demonstrate the effectiveness of their approach. The paper is well written, except for some minor typos listed below. It would be great if the authors can briefly mention a few real-life scenarios where one might expect to encounter gradient vectors that are sparse or approximately sparse. Typos: 1) Line 221: Figure 4 --> Figure 3 2) Line 232: Figure 4 --> Figure 3 3) Line 241: Figure 4.1 --> Figure 4 4) Line 277, 279: Figure 4.1 --> Figure 5. 5) The authors use $b_i$ to denote the labels. Please consider changing this notation as it conflicts with the number of bits $b$.

Reviewer 2



After rebuttal: In my opinion, this paper falls slightly short of the mark for NIPS, for the reasons outlined in my original review, but I believe it's worth communicating, and will not oppose its acceptance. I slightly upgraded my score since I agree to some extent that the algorithmic idea is not the same as QSGD; the ultimate decision lies with the area chair. ==== There's been recently a lot of work on the communication-variance trade-offs of distributed SGD. The present submission fits squarely into this space, by looking at the problem of sparsifying SGD gradients stochastically, to fit a certain variance budget. Thus, it aims to improve upon recent work like TernGrad and QSGD, which mainly try to reduce the communication budget by reducing bit-width (not necessarily sparsifying) gradients. The results can be summarized as follows: - The authors give an optimal scheme for fitting the sparsity in a gradient stochastically to a fixed variance budget. This is done by modeling the problem of setting choice probabilities for gradient entries in stochastic quantization as an optimization problem, for which the authors show that there is a nice closed-form solution. They then give an efficient way of implementing this. - The scheme is shown to provide both sparsity and variance bound guarantees. - The authors provide a relatively wide array of small benchmarks, for convex objectives (regression), non-convex (toy neural nets), and asynchronous parallel SGD. The experiments show that the scheme works well when compared with random and naive sampling approaches, and similarly (or a bit better) when compared with QSGD. My evaluation is as follows: On the positive side, the solution given here is reasonable, and appears correct to my reading. I think this is a nice set of results, which should be communicated in some form. On the negative side, I have serious doubts that this crosses the high bar set by NIPS. First, in terms of significance and improvement over previous work, the contribution here is, in my opinion, limited. The fact that stochastic quantization induces sparsity for low target bit width is not new. A basic version of this observation appears to be given in the QSGD paper: if we take their version with two quantization levels (0 and 1) it seems to me we get a version of this sparsification method with sqrt n density and sqrt n variance. The authors' contribution here is to set the question of choosing sampling probabilities efficiently given a specific variance budget as an optimization problem, which they solve efficiently. However, their optimal solution is quite similar to the QSGD scheme for a single non-zero quantization level. Thus, simplistically, it would seem to me like the contribution here is to determine the optimal scaling parameter for the sampling probabilities? Moreover, there are other related non-stochastic delay-based sparsification schemes, e.g. [1]. To my knowledge, these schemes do not have convergence guarantees, but they seem to ensure extremely high sparsity in practice for neural nets (an order of magnitude larger than what is considered here in the experimental section). Yet, there is minimal comparison with these, even just at the experimental level. Is the authors' claim that such schemes do not converge for convex objectives? This point is left ambiguous. Second, the experimental section is at the level of toy synthetic experiments, and therefore leaves much to be desired. I believe that validation on convex objectives is nice, and I haven't seen it in other papers in this area. But this is because communication-compression for regression problems is not really a critical performance problem these days, at least not at the scale shown here. The non-convex validation is done on toy CNNs, which would have no problem scaling on any decent setup. By contrast, the standard in the area is ImageNet or large-scale speech datasets, for which scaling is indeed a problem. Such experiments are not given here. The asynchronous experiment is more intriguing, but ultimately flawed. Basically, the idea here is that sparsity will help if threads compete on a bunch of memory locations (corresponding to the parameter). To boost contention, the authors *lock* each individual location (to my understanding). But why is this necessary(other than making the experiments look more impressive)? Modern computer architectures have compare&swap or fetch-and-add operations which could be used here, which wouldn't block access to those locations like locking does. These should be much faster in practice, and I believe standard analyses cover such asynchronous variants as well. Have the authors actually tried this? Another thing which the authors might want to keep in mind here is that, even though they ensure non-trivial sparsity (e.g. 70%) at each node, collectively the summed updates (gradients) may well become dense for reasonable node count. Hence this technique will not be very useful in any of the cases considered unless #dimension / #nodes >> sparsity. Summing up, while I have enjoyed reading this paper, I believe this is clearly below the threshold for NIPS, both on the theory and on the experimental side. Some minor comments: - sparsification technology -> "technique" is probably better here - you keep changing between [] and () for expectation - "The algorithm is much easier to implement, and computationally more efficient on parallel computing architecture." Why? - "after sparsified" -> please rephrase - "baseline of uniform sampling approach" -> please rephrase - Figure 2: overwriting the experimental results with the legend seems like a bad idea to me. - "converges much faster" -> what is "much" here? please be more precise - "employs a locked read" please be clearer with your experimental setup here. Are these reader-writer locks? phtread locks? test&set locks? The whole section should be re-written for clarity.

Reviewer 3



Summary: This paper presents a novel technique for reducing communication overheads in distributed optimization. Specifically, the method produces random sparsifications of stochastic gradients. Due to their sparsity, the gradients requires less communication cost. This work proposes an optimization problem that produces an unbiased gradient sparsification with a minimal number of non-zero entries, subject to some variance constraint. The authors derive a closed-form expression, as well as a greedy approximation algorithm that can be computed faster. Finally, the authors analyze the expected sparsity of the output of their method and empirical results demonstrating that their method can result in significantly reduced communication costs in practical settings. I believe that the paper is good, but needs some thorough revisions before submitting the final version. In particular, some of the theory needs more rigor and the paper could use revisions for spelling, grammar, and mathematical typesetting. Quality: The submission is generally sound. The proofs in the appendix are missing a few details and need some more elaboration, but can be made rigorous. Moreover, the empirical results are extensive and demonstrate the efficacy of the proposed method. The authors do a good job of discussing the pros and cons of their method, and discuss theoretically when the algorithm may be a good idea in Section 3. The authors make a few misleading claims about their work, however. In particular, they mention in the abstract that they introduce a convex optimization formulation. However, (3) is not clearly convex, so any claim to that effect needs justification. This issue is also in lines 38 and 293. The primary issue I have with the theory is that there is an occasional lack of rigor, especially the proof of Theorem 1. The authors apply the KKT conditions to the optimization problem in (3). However, (3) involves the condition that the probabilities p_i satisfy 0 < p_i <= 1. The KKT conditions generally apply only to inequalities, not strict inequalities. While I believe that almost all of their proof can go through, the proof as stated is not fully rigorous. Additionally, the authors apply complementary slackness in section 2.2 which again would need to be justified. Similarly, the use complementary slackness to determine that they want to find an index k satisfying (4). However, they claim that it follows that they should find the smallest index k satisfying this, which again needs more justification. One minor issue concerns Algorithm 3. While it is intended to be faster than Algorithm 2, it is not immediately clear that it is faster. In particular, Algorithm 2 requires O(klog(k)) computations, while each iteration of Algorithm 3 requires O(k) computations per iteration. Whether or not Algorithm 3 can be effective with fewer than log(k) iterations is not discussed. Moreover, the paper makes unsubstantiated claims that Algorithm 3 provides a good approximation to Algorithm 2 (as in Line 176-177). Either a theorem to this effect should be given or any such reference should be omitted or at least made more accurate. Clarity: The paper is generally easy to follow. However, there are a decent number of typos, grammatical issues, and mathematical typesetting issues that should be corrected. Some example issues are listed below: -Line 27: "2) about how to use" should just be "2) how to use" -Line 33-34: "with minor sacrifice on the number of iterations" is grammatically odd. Maybe something like "with only a minor increase in the number of iterations." -Line 34: "of our" should just be "our" -Line 49: The typesetting of x_n collides with the line above it -The second math equation after line 58 should be broken up -Line 59: Lipschitzian should just be Lipschitz -The spacing in the math line after 72 is inconsistent. -Equation (4) should be separated in to two lines, as the condition on lambda is distinct from the following inequality, which is subsequently referenced. -Line 172 "is relatively less increased comparing to the communication costs we saved" is grammatically awkward. -Line 237 "CIFAR10dataset" should be CIFAR-10 dataset. -The caption of Figure 4 intersects the text below it. Originality: The methods are, to the best of my knowledge, original. While the paper belongs to a growing body of literature on communication-efficient training methods, it fills a relatively unique role. Moreover, the paper does a good job of citing relevant papers. My only critique of the introduction is that descriptions of some of the most relevant work (such as [17] and [2]) are only briefly discussed, while some work (such as that on TernGrad, a competitor to QSGD) is omitted. Significance: This paper is an interesting, useful contribution to the growing work on communicatin-efficient distributed machine learning. It analyzes the task at a high level and as a result is applicable to any algorithm that involves communicating vectors in a distributed setting. Moreover, it takes the time to actually solve its associated optimization problem instead of simply proposing a method. Moreover, it does the necessary legwork to compare to many other related methods, both sparsification methods and quantization methods. Moreover, the work gives experiments on multiple algorithms, including SVRG. It is significant and hollistic in its approach. Again, the paper has a unique idea, good theory, and a good variety of experiments. Its only real flaws are a lack of rigor in some of the theory, some claims that are not fully substantiated (but whose omission does not detract significantly from the paper) and some grammatical and typesetting issues. These can all be remedied, and as long as this is done for the final version then I recommend acceptance.